ecology/evolution/behaviour

Bateman principles, reproductive success, mating system, anisogamy, rattlesnake, *Crotalus atrox*

**Author for correspondence:**
Brenna A. Levine
e-mail: levine.brenna.a@gmail.com

# No evidence of male-biased sexual selection in a snake with conventional Darwinian sex roles

Brenna A. Levine[1,2], Gordon W. Schuett[2,3], Rulon W. Clark[2,4], Roger A. Repp[5], Hans-Werner Herrmann[2,6] and Warren Booth[1,2]

[1]Department of Biological Science, The University of Tulsa, Tulsa, OK 74104, USA
[2]The Chiricahua Desert Museum, Rodeo, NM 88056, USA
[3]Department of Biology and Neuroscience Institute, Georgia State University, Atlanta, GA 30303, USA
[4]Department of Biology, San Diego State University, San Diego, CA 92182, USA
[5]National Optical Astronomy Observatory, Tucson, AZ 85719, USA
[6]School of Natural Resources and the Environment, University of Arizona, AZ 85721, USA

BAL, 0000-0002-4326-591X

Decades of research on sexual selection have demonstrated that 'conventional' Darwinian sex roles are common in species with anisogamous gametes, with those species often exhibiting male-biased sexual selection. Yet, mating system characteristics such as long-term sperm storage and polyandry have the capacity to disrupt this pattern. Here, these ideas were explored by quantifying sexual selection metrics for the western diamond-backed rattlesnake (*Crotalus atrox*). A significant standardized sexual selection gradient was not found for males ($\beta_{SS} = 0.588$, $p = 0.199$) or females ($\beta_{SS} = 0.151$, $p = 0.664$), and opportunities for sexual selection ($I_s$) and selection ($I$) did not differ between males ($I_s = 0.069$, $I = 0.360$) and females ($I_s = 0.284$, $I = 0.424$; both $p > 0.05$). Furthermore, the sexes did not differ in the maximum intensity of precopulatory sexual selection (males: $s'_{max} = 0.155$, females: $s'_{max} = 0.080$; $p > 0.05$). Finally, there was no evidence that male snout–vent length, a trait associated with mating advantage, is a target of sexual selection ($p > 0.05$). These results suggest a lack of male-biased sexual selection in this population. Mating system characteristics that could erode male-biased sexual selection, despite the presence of conventional Darwinian sex roles, are discussed.

# 1. Introduction

Bateman's [1] research on reproduction in fruit flies (*Drosophila melanogaster*) was first to experimentally test several key ideas of [2] theory on sex differences and sexual selection (reviewed in [3]). In the laboratory, Bateman compared the sexual behaviour and traits of male and female fruit flies and made the following three conclusions: (i) males have higher variance in the number of mates (i.e. mating success) than females, (ii) males have higher variance in the number of offspring produced (i.e. reproductive success) than females, and (iii) the slope of the relationship between mating and reproductive success is steeper in males. Although Bateman's original experiment has been rightly criticized for flawed methodology (e.g. [4]), these three conditions provide the conceptual framework by which sexual selection is measured in populations [3,5–7]. The slope of the regression that relates reproductive success to mating success (the degree to which the reproductive success increases with the number of mates obtained) is termed the sexual selection, or Bateman, gradient (*sensu* [3]). Accordingly, the sex which has the steeper Bateman gradient (typically the male) is predicted to experience the strongest selection pressure on traits that enhance mating success, such as body size, weaponry and/or specific behaviours [3]. By contrast, non-significant Bateman gradients imply lack of sexual selection on traits that are correlated with mating success [7].

As hypothesized by Bateman [1], differences in mating and reproductive success among males and females originate owing to differences in gametic size and investment (i.e. anisogamy; [8]). The female sex of most species only have a few (e.g. 1–100) large gametes (ova) relative to males, whereas males tend to have millions of much smaller gametes (spermatozoa). Consequently, females are likely to be more discriminating in their choices of mating partners, whereas males are more 'eager' to mate. With females as the limiting sex, anisogamy can ultimately: (i) result in greater variance in mating and reproductive success among males than females, given that males compete for mating opportunities with the often limited available females, (ii) promote a stronger relationship between mating success and reproductive success in males than females, and (iii) drive selection on traits that enhance male mating success [3,6,8,9].

However, Bateman's principles are also heavily affected by an extension of anisogamy, the operational sex ratio (OSR; the ratio of reproductive females to reproductive males; [10]). If female reproductive activity is brief and asynchronous, a male-biased OSR is expected, promoting competition among males for access to the few receptive females and increasing variance in mating and reproductive success among males [10]. Yet, other aspects of the mating system can weaken male-biased sexual selection. For example, multiple mating by females (i.e. polyandry) and the production of multiple litters/clutches per year can erode sexual selection in males by decreasing variance in male mating and reproductive success [11,12]. Nonetheless, sexual selection is generally male-biased across taxa with conventional Darwinian sex roles [13].

Snakes, in general, have emerged as models for patterns of male-biased sexual selection. Male-biased OSRs are common in snakes [14] due to energetic constraints on female reproduction [15]. Male competition for priority-of-access to females is frequent in many species (e.g. combat and mate guarding [15]), as is a resulting elevated variance in male mating and reproductive success compared to females [16–18]. Furthermore, snout–vent length (SVL), a trait often important in male–male combat (reviewed in [19]), has been found to be a target of sexual selection in some species [17].

However, other aspects of snake mating systems have the capacity to erode male-biased sexual selection. Most snake species investigated are considered to be polygynandrous or polyandrous [14]; females can have multiple mates within a season, and multiple paternity within litters is common [20], both of which can decrease variance in male mating and reproductive success. Long-term sperm storage that spans seasons, and even years, has also been documented [21,22] and has the potential to exacerbate this effect. Regardless, in most snakes investigated to date, sexual selection has been found to be greater in males than in females, even in systems with polyandry (e.g. northern water snake [16], black rat snake [23], copperhead [17], puff adder [18]).

In an effort to evaluate the contrasting effects of mating system characteristics on patterns of sexual selection, we quantified male and female sexual selection in a population of western diamond-backed rattlesnake (*Crotalus atrox*) using previously collected parentage and field data [24]. The mating system of *C. atrox* has the potential to promote male-biased sexual selection (e.g. male–male combat for access to females [25], male-biased sex ratio [24], male-biased sexual size dimorphism [26]), but also has characteristics that may contribute to its erosion (e.g. two mating seasons per year, multiple paternity [24] and long-term sperm storage [27]). We tested four hypotheses explicitly derived from traditional

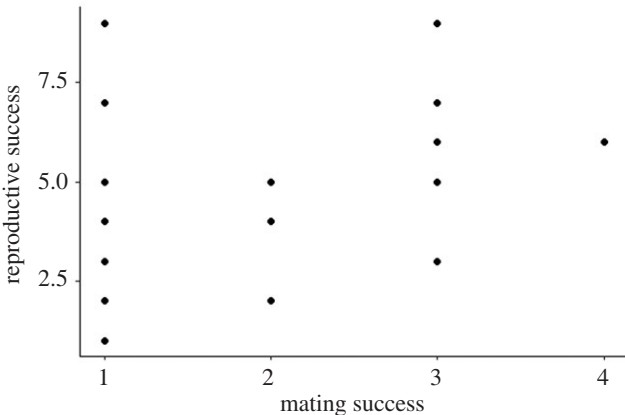

**Figure 1.** Reproductive success (= total number of offspring) with respect to mating success (= total number of mates) for male *C. atrox* (N = 27) studied in the Suizo Mountains (AZ, USA) from 2001 to 2010. There was no significant relationship between mating success and reproductive success for male *C. atrox* when accounting for the number of years that an individual bred ($\beta_{SS} = 0.303$, $p = 0.188$; standardized $\beta_{SS} = 0.588$, $p = 0.199$).

sexual selection theory: (H$_1$) males exhibit greater relative variance in mating success (i.e. opportunity for sexual selection) and reproductive success (i.e. opportunity for selection) than females; (H$_2$) males, but not females, exhibit a significant relationship between mating success and reproductive success (i.e. Bateman gradient); (H$_3$) males have a greater maximum intensity of precopulatory sexual selection than females (i.e. Jones index) and (H$_4$) males with longer SVL have significantly greater mating (i.e. mating differential) and reproductive success (i.e. selection differential).

# 2. Material and methods

## 2.1. Study system

Clark *et al.* [24] studied a single population of *C. atrox* in the Suizo Mountains (AZ, USA) over 10 consecutive years between 2001 and 2010. During this time, offspring (N = 108) were sampled from 18 known females. Following microsatellite genotyping, paternity was assigned to 18 sampled and 9 unsampled (i.e. not caught) males. Detailed information on the habitat, ecology, reproduction and mating system of this population can be found in Clark *et al.* [24], as can molecular and parentage analysis methods. The methods of Clark *et al.* [24] are also summarized in the electronic supplementary material.

Using these parentage assignments, the total mating and reproductive success for males and females over the course of the study were quantified (figures 1 and 2; electronic supplementary material), as were annual mating and reproductive success for males and females (electronic supplementary material). Here, mating success is defined as the number of mates with which an individual produced offspring, and reproductive success as the number of offspring produced by an individual, following Levine *et al.* [28].

## 2.2. Opportunities for sexual selection and selection

Opportunities for sexual selection ($I_S$) and selection ($I$) were quantified for each sex by year. For females, 5 years had sufficient data to calculate these metrics (greater than or equal to 3 records), whereas for males, 4 years yielded sufficient data. For each year, $I_S$ was calculated by dividing the sex-specific variance in mating success by the sex-specific squared mean of mating success. Similarly, annual $I$ was calculated by dividing the sex-specific variance in reproductive success by the sex-specific squared mean of reproductive success. These data were used to calculate the mean annual $I_S$ and $I$ for males and females. *F*-ratio tests were used to compare the mean annual $I$ and $I_S$ among males and females [29], with degrees of freedom estimated as 1 − mean number of annual records for males and females, respectively. Importantly, it should be noted that annual sample sizes are small, and this could affect the robustness of these calculations.

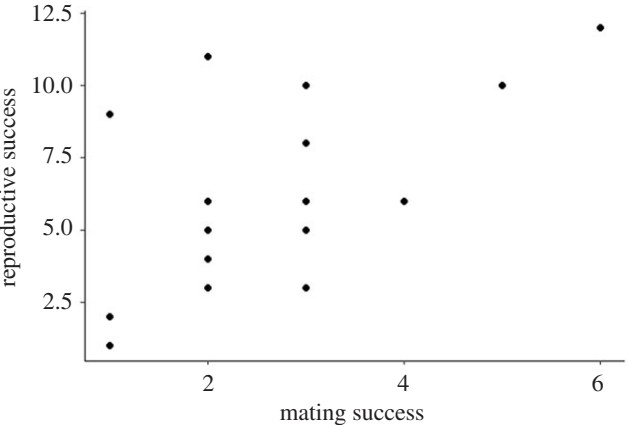

**Figure 2.** Reproductive success (= total number of offspring) with respect to mating success (= total number of mates) for female *C. atrox* ($N = 18$) studied in the Suizo Mountains (AZ, USA) from 2001 to 2010. There was no significant relationship between mating success and reproductive success for female *C. atrox* when accounting for the number of years that an individual bred ($\beta_{SS} = 0.045$, $p = 0.752$; standardized $\beta_{SS} = 0.151$, $p = 0.664$).

## 2.3. Bateman gradients

Absolute Bateman gradients ($\beta_{SS}$) were quantified using generalized linear models (GLMs) fit with the *glm{stats}* function in *RStudio* [30]. Sex-specific reproductive success was modelled as a function of sex-specific mating success, and models included as a covariate the number of years that an individual bred [31,32]. Initial models included an interaction between mating success and the number of breeding years; this interaction term was non-significant for both sexes and was subsequently removed from the models [33]. We also tested for collinearity between mating success and number of breeding years; because VIFs for males and females were less than 10 ([34]; male VIF = 5.502; female VIF = 5.415) and number of breeding years is also of biological importance in the model [35], we retained both predictor variables. A Poisson error distribution was employed for these GLMs.

Standardized $\beta_{SS}$ were also quantified, but using relative, rather than absolute, values of mating and reproductive success [7]. Relative mating and reproductive success were calculated for each individual by dividing the individual's mating and reproductive success by the sex-specific means of mating and reproductive success, respectively. As above, sex-specific relative reproductive success was modelled as a function of relative mating success, with the number of years that an individual bred included as a covariate. Gaussian error distributions were used for these models. For all models, we tested for a significant effect of mating success on reproductive success. This was done using likelihood ratio tests (LRTs) or tests of reduction in scaled deviance of each model versus a model lacking mating success as a predictor ($\chi^2$, $\alpha = 0.05$), as performed in *RStudio* using the *drop1{stats}* function. Finally, residual diagnostics of GLMs were evaluated using the *plot{graphics}* function in *RStudio*.

## 2.4. Jones index

The above metrics ($\beta_{SS}$ and $I_S$, in particular) have been criticized for yielding an incomplete representation of sexual selection due to a lack of integration of the relationship between mating and reproductive success ($\beta_{SS}$) with variance in mating success ($I_S$; [36]). This problem has been resolved by the development of the Jones index ($s'_{max}$), which represents the maximum intensity of precopulatory sexual selection by integrating $\beta_{SS}$ with $I_S$ [7], and which has recently been shown to outperform all other sexual selection metrics [36]. The Jones index was calculated for males and females by multiplying the sex-specific $\beta_{SS}$ by the square root of $I_S$ [7], with $I_S$ taken as the mean annual $I_S$ per sex. Male and female Jones indexes were statistically compared using an *F*-ratio test ($\alpha = 0.05$). Since $s'_{max}$ is in units of standard deviation [7], the squares of $s'_{max}$ for males and females were compared as above, with degrees of freedom estimated as $1 -$ mean number of annual records for males and females.

Of note, in calculations of $\beta_{SS}$, $I_S$, $I$ and $s'_{max}$, only mating and reproductive success of individuals that produced at least one offspring were considered (*sensu* [32]). Reproduction in female pitvipers is not constrained by mate availability (particularly when there is a male-biased sex ratio, as seen in this

study population [24]), but rather by energy allocation; females can only reproduce when they have sufficient physiological resources to do so [15]. Therefore, inclusion of females with zero reproductive success in $\beta_{SS}$ estimates would conflate variance in mating and reproductive success due to sexual selection with that due to energy availability.

For three reasons, analyses were also restricted to data for males that produced at least one offspring. First, in only analysing males that produced offspring, estimates of reproductive success for males and females were comparable, rather than inflated estimates of reproductive success for females when compared with males [31]. Second, this method allowed for the avoidance of a statistical dependency between mating and reproductive success: when mating success equals zero, reproductive success has to equal zero as well. Third, given the longitudinal nature of the data, this avoided a statistical interaction between mating success and the number of years that an individual bred, thus simplifying interpretation of the Bateman gradients. It should be noted, however, that excluding males with zero mating and reproductive success from analysis has its own potential to bias male $I_S$, $I$, $\beta_{SS}$ and $s'_{max}$, by underestimating male variance in mating and reproductive success [31].

An important caveat of these analyses is that mating success was inferred from genetic parentage assignments. Therefore, these data fail to capture mating events that did not result in production of offspring [37]. As a result, mating success values should be considered conservative estimates, with the potential to spuriously strengthen the relationship between mating and reproductive success [38].

## 2.5. Testing for selection on male SVL

Using the paternity inferences and field data of Clark *et al*. [24], annual mating and reproductive success were analysed for males that sired offspring and for which annual SVL estimates were available. These data comprised 27 annual records distributed among 18 males (electronic supplementary material). Relative mating and reproductive success were calculated for each male by dividing annual mating and reproductive success by mean annual mating and reproductive success, respectively [39]. Two metrics for SVL were then calculated: the linear mating differential ($m'$) and the linear selection differential ($s'$). These represent the relationships between male SVL and mating and reproductive success, respectively [36]. To estimate $m'$ for male SVL, annual relative mating success was regressed on to annual mean-standardized SVL. Male ID was included as a random effect in the model to account for repeated measures, as some males sired offspring in multiple years. Similarly, annual relative reproductive success was regressed on to annual mean-standardized SVL to calculate $s'$ for male SVL, also while controlling for repeated measures. Linear mixed-effects models were run in *RStudio* using the *lmer* function of package *lme4* [40]. The *drop1{stats}* function in *RStudio* was used to test for a significant effect of annual standardized SVL on annual relative mating and reproductive success with LRTs of the full models versus models lacking mean-standardized SVL as a predictor ($\chi^2$, $\alpha = 0.05$). All analyses in *RStudio* were accomplished using *R* v. 3.5.2 [41].

# 3. Results

## 3.1. Opportunities for sexual selection and selection

The mean annual $I_S$ for males was 0.069, whereas it was 0.284 for females. The mean annual $I_s$ for males and females did not differ significantly (*F*-ratio test, $p = 0.073$). The mean annual $I$ was 0.360 for males and 0.424 for females. Again, there was no significant difference between the mean annual $I$ for males and females (*F*-ratio test, $p = 0.430$).

## 3.2. Bateman gradients and Jones indexes

The effect of male mating success on male reproductive success was non-significant, regardless of the values analysed (absolute: $\beta_{SS} = 0.303$, $p = 0.188$; relative: standardized $\beta_{SS} = 0.588$, $p = 0.199$). Similarly, no significant effect of female mating success on female reproductive success was found when analysing absolute ($\beta_{SS} = 0.045$, $p = 0.752$) or relative values (standardized $\beta_{SS} = 0.151$, $p = 0.664$).

The Jones index ($s'_{max}$) integrates the sex-specific $\beta_{SS}$ with sex-specific $I_S$, thereby factoring the role of differential mating success into the relationship between mating and reproductive success and setting an upper limit on the strength of sexual selection acting on traits [36]. No significant difference in $s'_{max}$ for males ($s'_{max} = 0.155$) versus females ($s'_{max} = 0.080$; *F*-ratio test, $p = 0.089$) was found.

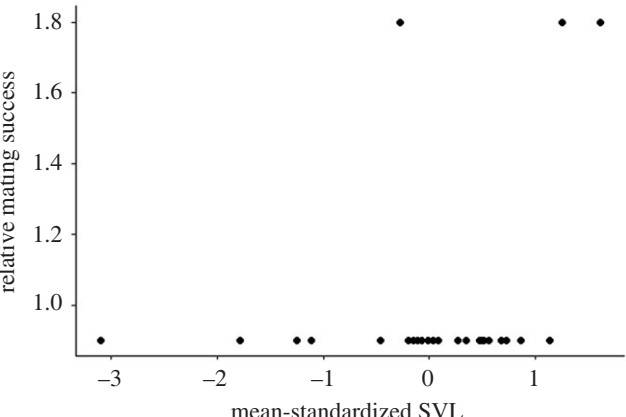

**Figure 3.** Annual relative mating success with respect to annual mean-standardized SVL for male *C. atrox* ($N = 18$) in the Suizo Mountains (AZ, USA). The mating differential ($m'$) for male SVL was not significant, as measured via a linear mixed-effects model in which annual relative mating success was regressed on annual mean-standardized SVL while accounting for repeated measures of males ($m' = 0.090$; $p = 0.096$).

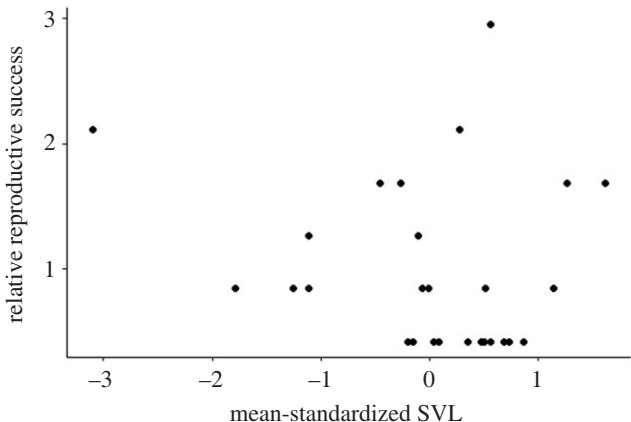

**Figure 4.** Annual relative reproductive success with respect to annual mean-standardized SVL for male *C. atrox* ($N = 18$) in the Suizo Mountains (AZ, USA). The linear selection differential ($s'$) for male SVL was not significant, as measured via a linear mixed-effects model in which annual relative reproductive success was regressed on annual mean-standardized SVL while accounting for repeated measures of males ($s' = 0.070$; $p = 0.697$).

## 3.3. Selection on male SVL

Annual male standardized SVL did not significantly affect annual male relative mating success ($m' = 0.090$; $p = 0.096$). This was unsurprising, given that, *within* a year, males tended to produce offspring with a single mate; only 3 of 27 annual records of male reproductive success reflected males producing offspring with two females (figure 3). Similarly, there was no significant effect of annual male standardized SVL on annual male relative reproductive success when accounting for repeated measures (figure 4; $s' = 0.070$, $p = 0.697$).

## 4. Discussion

Decades of theoretical and empirical research have investigated patterns of sexual selection (or lack thereof) in the wild, and a major theme has emerged from this research: males tend to experience greater sexual selection than females in systems with conventional Darwinian sex roles [13]. The aim of this study was to explore sexual selection in a species with mating system characteristics that could have contrasting effects on male-biased sexual selection. Although the mating system of this *C. atrox* population has characteristics that could decrease variance in male mating and reproductive success,

such as a polyandry [24], biannual mating seasons [42] and long-term sperm storage [24], male-biased sexual selection was expected in the study population for several reasons. First, the study population has a male-biased sex ratio (2:1; [24]) which should cause higher variance in mating and reproductive success among males than females and promote sexual selection in males [10,43]. Second, male–male combat for priority-of-access to females occurs in this species [24,25], and this should both contribute to elevated variance in male mating and reproductive success [44] and drive sexual selection on traits that increase performance in these combat bouts (e.g. larger SVL) [45]. Finally, *C. atrox* exhibit male-biased sexual size dimorphism [26], a common consequence of male–male combat exerting selection pressure on male body size [46]. Nevertheless, no evidence of male sexual selection was found in the study population when quantifying multiple metrics of sexual selection.

In contrast with this study system, male-biased sexual selection has been found across a variety of other snake taxa, including those with mating systems characteristics that could similarly weaken male sexual selection (e.g. northern water snake, *Nerodia sipedon* [16]; black rat snake, *Pantherophis obsoletus* [23]; copperhead, *Agkistrodon contortrix* [17], puff adder, *Bitis arietans* [18]), yet these species also exhibit multiple mating by females and clutches/litters with multiple paternity. However, one distinct difference between these systems and that described here is the number of mating seasons experienced per year. This *C. atrox* population exhibits two mating seasons per year [24], as opposed to one mating season per year in the study populations of these other snake species. It is, therefore, possible that the additional mating opportunities afforded to males by a second mating season decreases variance in male mating and reproductive success to such an extent that male-biased sexual selection is eroded. This hypothesis should be tested through comparative studies of species with populations that can exhibit one or two mating seasons per year (e.g. copperheads; [47]).

A likely contributing factor to the apparent lack of male-biased sexual selection in this study population is long-term sperm storage. Long-term sperm storage occurs in a variety of snake taxa [22], including *C. atrox* [24]. Unlike mammals, which are generally unable to store sperm for longer than 24 h [48], reptiles have been shown to store sperm not only across the duration of a breeding season, but also across multiple seasons [49–51]. In fact, the longest, genetically confirmed, case of long-term sperm storage comes from the eastern diamond-backed rattlesnake, *C. adamanteus*, which produced a litter of 19 neonates, 67 months after being field-collected and housed in captive isolation until the birth [21]. For females, the ability to store sperm offers many reproductive advantages to include: decoupling the timing of mating from ovulation [15], enabling sperm competition [52,53], escape from inbreeding [54] and the maximization of genetic diversity within and across litters [55]. As such, it is likely that long-term sperm storage is common, at least within a given breeding season, and potentially across multiple seasons, within the study population. Female sperm storage also has the potential to influence the strength of male sexual selection. The removal of the need for post-copulation mate guarding, given that it provides little assurance of full paternity of the resulting offspring, affords males greater opportunity to seek additional mates. Furthermore, if females mix sperm following matings with multiple males, winning competitions for priority access to females (i.e. combat dances) is less likely to lead to higher male reproductive success. An assessment of the duration and significance of long-term sperm storage in this study population of *C. atrox* would clarify its relationship to the pattern of sexual selection observed here.

While it is likely that the apparent lack of male-biased sexual selection found here results from characteristics of the mating system of the study population, a non-biological explanation could be that only males with non-zero reproductive success were analysed. As such, this method has the capacity to underestimate male variance in mating and reproductive success [31]. Indeed, if males with zero mating and reproductive success are included in analyses, a significant relationship between male mating and reproductive success emerges (electronic supplementary material). However, in doing so, the number of breeding years and the interaction of this covariate with mating success become significant as well, such that the effect of male mating success on reproductive success cannot be interpreted without also considering its interaction with the number of breeding years.

However, it is unlikely that the exclusion of males with zero mating and reproductive success from analyses is responsible for the lack of evidence of male-biased sexual selection observed in the study population for three reasons. First, mating and selection differentials on male SVL, a trait that is the target of sexual selection in other snakes with male–male combat for access to females [17], were not significant. Complementing these results, Clark *et al.* [24] previously analysed the SVLs of males that did and did not produce offspring in the *C. atrox* population, and found no difference in SVLs among the groups. A lack of selection on male SVL is further supported by less pronounced sexual

size dimorphism in the study population, when compared with other *C. atrox* populations [26,56]. Indeed, male-biased sexual size dimorphism is a manifestation of male-biased sexual selection and conventional sex roles [13]. Thus, even though sexual size dimorphism is present in the study population, a lesser degree of it compared to other populations combined with a non-significant relationship between male SVL and mating or reproductive success is indicative of a true lack of male-biased sexual selection.

Second, in an effort to evaluate the impact of including or excluding males with zero mating and reproductive success on the detection of male-biased sexual selection, a published dataset from a study of sexual selection in copperhead snakes, *A. contortrix* [17], was reanalysed. In this study, the authors included males with zero mating and reproductive success in analyses and found male-biased sexual selection. Re-analysis of these data excluding males with zero mating and reproductive success found that there was no longer a significant difference between male and female $I_s$ ($p > 0.05$; electronic supplementary material). However, the significant difference between male and female $I$ remained, and the male ($p < 0.01$), but not female ($p > 0.05$), $\beta_{SS}$ was still statistically significant. Although not estimated by Levine *et al.* [17], the standardized $\beta_{SS}$ and $s'_{max}$ was quantified for males and females when excluding males with zero reproductive success. No significant standardized $\beta_{SS}$ was detected for females ($p > 0.05$), but a significant standardized $\beta_{SS}$ still resulted for males ($p = 0.05$). Most importantly, $s'_{max}$, a measure recently found to most closely approximate the actual strength of sexual selection in a population [36], was significantly greater for male copperheads than female copperheads (male $s'_{max}$: 0.260, female $s'_{max}$: 0.063; $p < 0.01$; electronic supplementary material) with data filtered to exclude males with zero mating and reproductive success. Thus, multiple lines of evidence ($I$, $\beta_{SS}$, standardized $\beta_{SS}$ and $s'_{max}$) support male-biased sexual selection in *A. contortrix* when only males with non-zero mating and reproductive success are considered. These results are further supported by a significant selection gradient on male SVL in *A. contortrix* [17].

Third, mating success in this study was inferred from genetic parentage assignments, and, therefore, failed to capture mating events that did not result in reproductive success [37]. Estimation of mating success from reproductive success in this way is anti-conservative, having the potential to erroneously strengthen, rather than weaken, the relationship between mating and reproductive success [38]. Despite this less conservative method, male $\beta_{SS}$ in *C. atrox* was still not significant.

Finally, it should be noted that the small sample size for males in this study could be responsible for the failure to detect significant sexual selection in *C. atrox*. However, this is unlikely for two reasons. First, and as above, genetic parentage assignments tend to upward bias estimates of sexual selection by resulting in a lower bound for data points [31]. Second, the male sample sizes in this study for both estimation of $\beta_{SS}$ and selection on male SVL exceed those in the re-analysis of the male *A. contortrix* data of Levine *et al.* [17]. Nevertheless, male *A. contortrix* experienced both significant $\beta_{SS}$ and selection on male SVL, whereas we found no evidence that male *C. atrox* did so.

In conclusion, empirical evidence supports a lack of male-biased sexual selection in a species for which, based on sexual selection theory and conventional Darwinian sex roles, it seemingly should exist. Although future research will be needed to determine why this population does not exhibit male-biased sexual selection, characteristics of the mating system that contribute to multiple mating by females and the potential for long-term sperm storage are likely culprits for erosion of male-biased sexual selection in *C. atrox*.

Ethics. The Institutional Animal Care and Use Committee (IACUC) of Arizona State University approved this study (protocol 98-429R). Other appropriate scientific permits were obtained from the Arizona Game and Fish Department (see Clark *et al.* [24]).

Data accessibility. All data and R code are available as electronic supplementary material. Data and relevant code for this research work are also stored in GitHub: https://github.com/brenna-levine/W_Diamondback_Sexual_Selection and have been archived within the Zenodo repository: https://doi.org/10.5281/zenodo.3997724.

Authors' contributions. G.W.S. and R.W.C. designed the original *C. atrox* study. G.W.S. and R.A.R. collected data. R.W.C. assembled the long-term data. R.W.C. and H.-W.H. analysed the genotypic data for the original study. B.A.L. performed the statistical analyses. B.A.L., G.W.S. and W.B. wrote the paper. All authors revised and approved the final version of the manuscript.

Competing interests. The authors declare no competing interests.

Funding. Funding was by a Research Incentive Award (US National Science Foundation) and a Research Creative Activities Award (Arizona State University West) to G.W.S. Arizona State University, Zoo Atlanta, Georgia State University, San Diego State University, The University of Tulsa and David L. Hardy Sr contributed funding support.

Acknowledgements. We greatly benefitted by discussions with many colleagues, especially Adam Jones, Steve Arnold, Harry Greene, Randall Reiserer and Emily Taylor.

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
