## [Reviewer comments · Royal Society Open Science]

Review History

RSOS-201261.R0 (Original submission)

Review form: Reviewer 1

Is the manuscript scientifically sound in its present form?

Yes

Are the interpretations and conclusions justified by the results?

No

Is the language acceptable?

Yes

Do you have any ethical concerns with this paper?

No

Have you any concerns about statistical analyses in this paper?

No

Recommendation?

Accept with minor revision (please list in comments)

Comments to the Author(s)

Comments to Authors on RSOS-201261 – Levine et al. “No evidence of male-biased sexual selection in a snake with conventional Darwinian sex roles”

With two exceptions (specified below) we think the authors have done an adequate job addressing our comments on the original manuscript submitted to Proceedings of Roy Soc. B.

Comment #2 – The end of the authors’ response to this comment “Therefore, we opt not to provide additional potential traits that could be under sexual selection” might be interpreted to mean that they do have data on other traits, but choose not to include them. We do not think this is the case, but please clarify. We are not suggesting that collection of data on additional traits is required if they are not available. However, if the analysis remains limited to only SVL, then it also follows that it is only valid to conclude there was no sexual selection on SVL in male and female *C. atrox*. Drawing the conclusion that that sexual selection is absent entirely in this population simply extends the interpretation beyond the data that is presented. Accordingly, such statements should be modified to “no evidence of sexual selection on SVL”, including in the manuscript title. Publishing the manuscript with the current title runs the risk of others not investigating other traits in this snake.

Comment about analyzing other characters - The authors are of course correct that selection may go undetected for many reasons. But this response misses our point, and we suggest that the authors are also ignoring an opportunity. Potential targets of sexual selection should be chosen for investigation based on biological details that might pertain to the taxon of interest. For example, sexual selection studies in lizards often involve measurements of one or more head dimensions and the lengths of tails and limbs in addition to total size (SVL, mass) because these metrics have been shown to scale allometrically in some species, and may play a role in sexually-selected behavior patterns and performance traits (e.g., copulation, running speed). The fact that studies on other Viperid snakes have found sexual selection on SVL does not eliminate the possibility that it may also act on other traits. We are simply suggesting that as an alternate explanation of their negative finding, the authors expand on the point that it could be worthwhile for snake researchers to examine other potential targets of sexual selection, even if they do not include traits other than SVL in this paper. We do appreciate that there is limited room to speculate given the severe length restrictions of the journal, but perhaps there is room for a brief mention (one sentence) in the Discussion about other traits like those that we speculated about. Who knows? - doing so might start a new trend that reveals unexpected findings.

L118-120. We do not agree that summarizing the information presented in Clark et al., (2014) that is pertinent to the message of the present study would “muddy” the message of the manuscript. If anything, a short summary in the supplement that readers can consult without searching for Clark et al. (2014) would make the message more accessible for readers. We would really think adding such information supplementally, would improve the manuscript, and it can be done without much effort.

Decision letter (RSOS-201261.R0)

Dear Dr Levine

On behalf of the Editors, we are pleased to inform you that your Manuscript RSOS-201261 "No Evidence of Male-Biased Sexual Selection in a Snake with Conventional Darwinian Sex Roles" has been accepted for publication in Royal Society Open Science subject to minor revision in accordance with the referees' reports. Please find the referees' comments along with any feedback from the Editors below my signature.

Please submit your revised manuscript and required files (see below) no later than 7 days from today's (ie 20-Aug-2020) date. Note: the ScholarOne system will 'lock' if submission of the revision is attempted 7 or more days after the deadline. If you do not think you will be able to meet this deadline please contact the editorial office immediately.

on behalf of Dr Alexander Ophir (Associate Editor) and Kevin Padian (Subject Editor)
openscience@royalsociety.org

Associate Editor Comments to Author (Dr Alexander Ophir):

Comments to the Author:

Dear Dr. Levine,

I have been assigned your manuscript after transfer from Proc Roy Soc B, and I received your revised draft and the comments from one of your original reviewers of your revised draft. Their comments are appended below.

As you will see, they felt that your comments sufficiently addressed their original concerns. After review of the manuscript, the reviewer's comments, and your responses, I agree that these are satisfactory. However, the reviewer raised two points that I believe you should address before proceeding to publication. The first comment about providing a somewhat more conservative conclusion based on SVL and avoiding potential overgeneralization is important and should be easily addressed. Similarly, providing a fair discussion of Clark et al. 2014 is justified and without the word constraints required of Proc Roy Soc B, you should be able to rather easily add this in to provide more context for the readers' benefit.

On a personal note, I quite enjoyed reading this manuscript and appreciated that it challenges traditional dogma within the area of sexual selection. I will be excited to see it in the literature once you have addressed these two points.

Alex Ophir
Associate Editor, RSOS

Reviewer comments to Author:
Reviewer: 1

Comments to the Author(s)

Comments to Authors on RSOS-201261 – Levine et al. “No evidence of male-biased sexual selection in a snake with conventional Darwinian sex roles”

With two exceptions (specified below) we think the authors have done an adequate job addressing our comments on the original manuscript submitted to Proceedings of Roy Soc. B.

Comment #2 – The end of the authors’ response to this comment “Therefore, we opt not to provide additional potential traits that could be under sexual selection” might be interpreted to mean that they do have data on other traits, but choose not to include them. We do not think this is the case, but please clarify. We are not suggesting that collection of data on additional traits is required if they are not available. However, if the analysis remains limited to only SVL, then it also follows that it is only valid to conclude there was no sexual selection on SVL in male and female *C. atrox*. Drawing the conclusion that that sexual selection is absent entirely in this population simply extends the interpretation beyond the data that is presented. Accordingly, such statements should be modified to “no evidence of sexual selection on SVL”, including in the manuscript title. Publishing the manuscript with the current title runs the risk of others not investigating other traits in this snake.

Comment about analyzing other characters - The authors are of course correct that selection may go undetected for many reasons. But this response misses our point, and we suggest that the authors are also ignoring an opportunity. Potential targets of sexual selection should be chosen for investigation based on biological details that might pertain to the taxon of interest. For example, sexual selection studies in lizards often involve measurements of one or more head dimensions and the lengths of tails and limbs in addition to total size (SVL, mass) because these metrics have been shown to scale allometrically in some species, and may play a role in sexually-selected behavior patterns and performance traits (e.g., copulation, running speed). The fact that studies on other Viperid snakes have found sexual selection on SVL does not eliminate the possibility that it may also act on other traits. We are simply suggesting that as an alternate explanation of their negative finding, the authors expand on the point that it could be worthwhile for snake researchers to examine other potential targets of sexual selection, even if they do not include traits other than SVL in this paper. We do appreciate that there is limited room to speculate given the severe length restrictions of the journal, but perhaps there is room for a brief mention (one sentence) in the Discussion about other traits like those that we speculated about. Who knows? - doing so might start a new trend that reveals unexpected findings.

L118-120. We do not agree that summarizing the information presented in Clark et al., (2014) that is pertinent to the message of the present study would “muddy” the message of the manuscript. If anything, a short summary in the supplement that readers can consult without searching for Clark et al. (2014) would make the message more accessible for readers. We would really think adding such information supplementally, would improve the manuscript, and it can be done without much effort.

===PREPARING YOUR MANUSCRIPT===

===PREPARING YOUR REVISION IN SCHOLARONE===

- If you are providing image files for potential cover images, please upload these at this step, and inform the editorial office you have done so. You must hold the copyright to any image provided.
- A copy of your point-by-point response to referees and Editors. This will expedite the preparation of your proof.

- Ensure that your data access statement meets the requirements at <https://royalsociety.org/journals/authors/author-guidelines/#data>. You should ensure that you cite the dataset in your reference list. If you have deposited data etc in the Dryad repository, please only include the 'For publication' link at this stage. You should remove the 'For review' link.
- If you are requesting an article processing charge waiver, you must select the relevant waiver option (if requesting a discretionary waiver, the form should have been uploaded at Step 3 'File upload' above).
- If you have uploaded ESM files, please ensure you follow the guidance at <https://royalsociety.org/journals/authors/author-guidelines/#supplementary-material> to include a suitable title and informative caption. An example of appropriate titling and captioning may be found at https://figshare.com/articles/Table_S2_from_Is_there_a_trade-off_between_peak_performance_and_performance_breadth_across_temperatures_for_aerobic_scop_e_in_teleost_fishes_/3843624.

Author's Response to Decision Letter for (RSOS-201261.R0)

See Appendix A.

RSOS-201261.R1 (Revision)

Review form: Reviewer 1

Is the manuscript scientifically sound in its present form?

Yes

Are the interpretations and conclusions justified by the results?

Yes

Is the language acceptable?

Yes

Do you have any ethical concerns with this paper?

No

Have you any concerns about statistical analyses in this paper?

No

Recommendation?

Accept as is

Comments to the Author(s)

The changes and author explanations in response to our comments are adequate.

Decision letter (RSOS-201261.R1)

Dear Dr Levine,

It is a pleasure to accept your manuscript entitled "No Evidence of Male-Biased Sexual Selection in a Snake with Conventional Darwinian Sex Roles" in its current form for publication in Royal Society Open Science. The comments of the reviewer(s) who reviewed your manuscript are included at the foot of this letter.

You can expect to receive a proof of your article in the near future. Please contact the editorial office (opencience_proofs@royalsociety.org) and the production office (opencience@royalsociety.org) to let us know if you are likely to be away from e-mail contact -- if you are going to be away, please nominate a co-author (if available) to manage the proofing process, and ensure they are copied into your email to the journal.

Best regards,
Lianne Parkhouse
Editorial Coordinator
Royal Society Open Science
opencience@royalsociety.org

on behalf of Professor Kevin Padian (Subject Editor)
opencience@royalsociety.org

Reviewer comments to Author:

Reviewer: 1

Comments to the Author(s)

The changes and author explanations in response to our comments are adequate.

Appendix A

RESPONSE TO REVIEWER COMMENTS

NO EVIDENCE OF MALE-BIASED SEXUAL SELECTION IN A SNAKE WITH CONVENTIONAL DARWINIAN SEX ROLES

Brenna A. Levine^{1,2}, Gordon W. Schuett^{2,3}, Rulon W. Clark,^{2,4} Roger A. Repp⁵, Hans-
Werner Herrmann^{2,6}, and Warren Booth^{1,2}

¹ Department of Biological Science, The University of Tulsa, Tulsa, OK 74104, USA

² The Chiricahua Desert Museum, Rodeo, NM 88056, USA

³ Department of Biology and Neuroscience Institute, Georgia State University, Atlanta,
GA 30303, USA

⁴ Department of Biology, San Diego State University, San Diego, CA 92182, USA.

⁵ National Optical Astronomy Observatory, Tucson, AZ 85719, USA

⁶ School of Natural Resources and the Environment, University of Arizona, AZ 85721,
USA

Corresponding author: Brenna A. Levine (levine.brenna.a@gmail.com)

Keywords: Bateman principles, reproductive success, mating system, anisogamy,
rattlesnake, *Crotalus atrox*

We thank the reviewer and associate editor of *Royal Society Open Science* for their careful reading and critique of both our revision and response to previous reviewer comments. Below, you will find our responses (*red, italicized text*) to reviewer comments (**bold**).

The end of the authors' response to this comment "Therefore, we opt not to provide additional potential traits that could be under sexual selection" might be interpreted to mean that they do have data on other traits, but choose not to include them. We do not think this is the case, but please clarify. We are not suggesting that collection of data on additional traits is required if they are not available. However, if the analysis remains limited to only SVL, then it also follows that it is only valid to conclude there was no sexual selection on SVL in male and female *C. atrox*. Drawing the conclusion that that sexual selection is absent entirely in this population simply extends the interpretation beyond the data that is presented. Accordingly, such statements should be modified to "no evidence of sexual selection on SVL", including in the manuscript title. Publishing the manuscript with the current title runs the risk of others not investigating other traits in this snake.

*The reviewer is correct that we do not have data on other traits, and we apologize for the confusion in our previous response. Regarding the analysis of sexual selection on SVL, we agree that this particular analysis can only conclude that there was no sexual selection on SVL in male *C. atrox*, and cannot be extended to conclusions regarding other traits. However, we also quantified sexual selection gradients (i.e., Bateman gradients) for male and female *C. atrox*, and these metrics of sexual selection are distinct from phenotypic selection gradients on mating success. Importantly, non-significant Bateman gradients, such as those reported here, imply that traits associated with mating success experience weak or no sexual selection (Jones et al. 2009). Therefore, our data do not just demonstrate that there is a lack of sexual selection acting on SVL, but also suggest that sexual selection is limited or entirely absent for any trait related to male or female mating success in *C. atrox*. As such, we respectfully disagree with the reviewer that our interpretation extends beyond the data. For this reason, it would be incorrect to change the title of our manuscript to include "on SVL", as this would not cover the other results of the manuscript.*

*However, we hear the reviewer's concern and have added some clarifying language throughout the manuscript to both assure the reader that we are not over-interpreting our results and to emphasize that we found no evidence of significant sexual selection in *C. atrox* (as opposed to stating that it does not exist, definitively). We*

also note the presence of such language throughout the manuscript where it currently exists.

Locations of these changes and original language in the track-changed document:

Line 1: title includes “no evidence of sexual selection”

Lines 33-35: Modified sentences to read “These results suggest a lack of male-biased sexual selection in this population of *C. atrox*. Mating system characteristics that could erode male-biased sexual selection, despite the presence of conventional Darwinian sex roles, are discussed.”

Lines 60-62: Added sentence that reads “In contrast, non-significant Bateman gradients imply of lack of sexual selection on traits that are correlated with mating success (Jones, 2009).” This sentence should explain to the reader that our Bateman gradient results suggest that there is no sexual selection in this population (as above).

Lines 266 – 267: sentence already included “no evidence”

Lines 283 and 306: added the word “apparent” before the phrase “lack of sexual selection”

Line 317: added the words “evidence of” so phrase now reads “lack of evidence of sexual selection”

Lines 362-363: modified sentence to read “Nevertheless, male *A. contortrix* experienced both significant β_{SS} and selection on male SVL, whereas we found no evidence that male *C. atrox* did so.”

Line 364: already read “empirical evidence supports a lack of male-biased sexual selection”

The authors are of course correct that selection may go undetected for many reasons. But this response misses our point, and we suggest that the authors are also ignoring an opportunity. Potential targets of sexual selection should be chosen for investigation based on biological details that might pertain to the taxon of interest. For example, sexual selection studies in lizards often involve measurements of one or more head dimensions and the lengths of tails and limbs in addition to total size (SVL, mass) because these metrics have been shown to scale allometrically in some species, and may play a role in sexually-selected behavior patterns and performance traits (e.g., copulation, running speed). The fact that studies on other Viperid snakes have found sexual selection on SVL does not eliminate the possibility that it may also act on other traits. We are simply suggesting that as an alternate explanation of their negative finding, the authors expand on the point that it could be worthwhile for snake researchers to examine other potential targets of sexual selection, even if they do not include traits other than SVL in this paper. We do appreciate that there is limited room to speculate given the severe length restrictions of the journal, but perhaps there is

room for a brief mention (one sentence) in the Discussion about other traits like those that we speculated about. Who knows? - doing so might start a new trend that reveals unexpected findings.

*As explained in our response to the first comment, the results of our Bateman gradient analysis imply that there is no significant sexual selection acting on traits related to mating success in male *C. atrox*. It would be thus be odd for us to suggest other traits that could be under sexual selection in male *C. atrox*, considering our non-significant Bateman gradient implies that there are none.*

L118-120. We do not agree that summarizing the information presented in Clark et al., (2014) that is pertinent to the message of the present study would “muddy” the message of the manuscript. If anything, a short summary in the supplement that readers can consult without searching for Clark et al. (2014) would make the message more accessible for readers. We would really think adding such information supplementally, would improve the manuscript, and it can be done without much effort.

We have added a section to the supplemental material summarizing the methods of Clark et al. (2014). Please see the new section: “3. Summary of Clark et al. (2014)”. We have also added the following statement to lines 125 – 126: “The methods of Clark et al. (2014) are also summarized in the supplemental material.”

References:

*Jones AG. 2009 On the opportunity for sexual selection, the Bateman gradients, and the maximum intensity of sexual selection. *Evolution* **63**, 1673–1684. (doi:10.1111/j.1558-5646.2009.00664.x)*